# PINNferring the Hubble Function with Uncertainties

Lennart Röver [1], Björn Malte Schäfer [1,3], and Tilman Plehn[2,3]

**1** Zentrum für Astronomie der Universität Heidelberg, Astronomisches Rechen-Institut, Heidelberg, Germany
**2** Institut für Theoretische Physik, Universität Heidelberg, Germany
**3** Interdisciplinary Center for Scientific Computing (IWR), Universität Heidelberg, Germany

## Abstract

The Hubble function characterizes a given Friedmann-Robertson-Walker spacetime and can be related to the densities of the cosmological fluids and their equations of state. We show how physics-informed neural networks (PINNs) emulate this dynamical system and provide fast predictions of the luminosity distance for a given choice of densities and equations of state, as needed for the analysis of supernova data. We use this emulator to perform a model-independent and parameter-free reconstruction of the Hubble function on the basis of supernova data. As part of this study, we develop and validate an uncertainty treatment for PINNs using a heteroscedastic loss and repulsive ensembles.

# 1 PINNtroduction

Friedmann-Robertson-Walker spacetimes are entirely characterized by their Hubble function $H(a) = \dot{a}/a$ as a consequence of the cosmological principle, which requires homogeneity and isotropy on cosmological scales. Substitution of the Friedmann-Robertson-Walker line element into the gravitational field equation allows to relate the Hubble function to the densities of the cosmological fluids and their respective equations of state. Probing the Hubble function not only addresses these densities and their equations of state, but also interactions between the fluids and their possible non-adiabatic evolution. The Hubble function itself only relies on symmetry assumptions and is in fact not restricted by theory to follow any a-priori parameterization. As such, it is possible to reconstruct it without recourse to a specific cosmological model, like assumptions about the gravitational field equation or specific properties of the cosmological fluids.

Perhaps the most direct probe of cosmic evolution out to redshifts beyond unity are supernovae of type Ia [1, 2]. They allow constraints on the evolution of luminosity distance with redshift, and therefore indirectly on the Hubble function, from which the luminosity distance follows after an integration. A typical effect indicative of repulsive gravity on large scales are systematically darker supernovae as they approach the cosmic horizon. These effects are associated with cosmological fluids with equations of state $w < -1/3$, typical for dark energy or the cosmological constant. In many cases, the equation of state is an immutable property of the cosmic fluid, for instance $w = 0$ for matter, and often one works with constant or linearly evolving equations of state for dark energy [3, 4], even though there are compelling arguments for evolving dark energy [5–9].

Our approach relies only on the symmetry principles for spacetime and derives the Hubble function $H(a)$ free of any parameterization directly from data. We achieve this with physics-informed neural networks (PINNs) [10–15], similar techniques have been used in Refs. [16–20]. They absorb the space of solutions of a differential equation with a free representation of the Hubble function given by a second neural network. This is necessary because the Hubble function is not directly observable. The relevant observable is the luminosity distance, a weighted integral over the inverse Hubble function. The PINN needs to learn a fast prediction of the luminosity distance for a given Hubble function, which is represented by another neural network.

In physics, a number without an error bar cannot describe a measurement and it also cannot describe a prediction. This means, we need to develop an uncertainty estimate for the learned function encoded in the PINNs. Sources of uncertainties of a trained neural network include statistical limitation of the training sample, stochasticity or noise of the training data, theory uncertainties in simulated training data, or a lack of flexibility of the network architecture [21]. We employ two different methods to learn an error band on the network prediction, a heteroscedastic loss function [22, 23] and repulsive ensembles [24].

Our paper starts with a brief introduction to PINNs and the two methods to also learn an uncertainty in Sec. 2. To the best of our knowledge, we apply repulsive ensembles to a cosmological problem for the first time, so we include a more detailed derivation in Sec. 2.3. In Sec. 3 we train a PINN emulator and study its behavior. Finally, we show how the combination of two networks can be used to extract the Hubble function from the luminosity in Sec. 4. We find that our inference works great, up to redshifts where the experimental uncertainties do not allow for a meaningful extraction anymore.

## 2 PINNcertainties

The idea behind physics-informed neural networks [10–15] is to understand and reproduce training data more efficiently by learning it as a solution to a differential equation. For an ordinary differential equation,

$$\dot{u}(t) = F(u, t) \qquad \text{with initial conditions} \qquad u(t = 0) = u_0 \,, \tag{1}$$

their MSE loss consists of two terms,

$$\begin{aligned}
\mathcal{L} &= (1 - \beta)\mathcal{L}_{\text{IC}} + \beta\mathcal{L}_{\text{ODE}} \\
\text{with} \qquad \mathcal{L}_{\text{IC}} &= [u_\theta(t = 0) - u_0]^2 \\
\mathcal{L}_{\text{ODE}} &= [\dot{u}_\theta(t) - F(u_\theta, t)]^2 \,.
\end{aligned} \tag{2}$$

PINNs form, together with neural differential equations and neural operators, a group of machine learning methods relating neural networks to solutions of differential equations. Here, PINNs learn a prediction for a given parameter choice without really solving an ODE at the stage of evaluation. Neural ODEs [25] use neural networks as part of a system of differential equations that is solved with conventional methods. Neural operators [26] provide a parameterized mapping of e.g. initial conditions to a state at a given time, but can be used in a more general context.

The first term drives the PINN to fulfill the initial conditions, and can be used without any additional training data. The second term ensures that the network approximates a solution to the differential equation. The parameter $\beta$ balances the two contributions.

The PINN training through the ODE loss uses two kinds of data. First, unlabeled or residual data points consist of points in time, where the differential equation is evaluated during the training [27]. For the ODE loss these time points determine where the differential equation is evaluated. Second, labeled time points can include other information, in our case the corresponding true values for $u(t)$ and $\dot{u}(t)$.

### 2.1 Toy example

We demonstrate some properties of PINNs for a simple harmonic oscillation. This includes the quality of the approximation for an increasing number of residual points, the effect of including labeled data, and how to estimate uncertainties using a heteroscedastic loss and repulsive ensembles. Our toy model is defined by the differential equation in two dimensions,

$$\ddot{u} + \frac{u}{2} = 0 \qquad \text{with} \qquad u(0) = \begin{pmatrix} 1 \\ 0 \end{pmatrix} \qquad \dot{u}(0) = \begin{pmatrix} 0 \\ 1 \end{pmatrix}. \tag{3}$$

The PINNs are trained on a first-order ODE describing the evolution of the vector $(u, \dot{u})$, where the two dimensions are independent of each other. This has the advantage of slightly faster training, but it sacrifices the guaranteed relation between $u$ and $\dot{u}$. For all results, we show one of the two components $u_{1,2}(t)$.

As a complication, our harmonic oscillator, Eq.(3), has a trivial solution $u(t) = 0$. For this solution $\mathcal{L}_{\text{IC}}$ is not minimal, but $\mathcal{L}_{\text{ODE}}$ does not lead to any gradient. Only the coupled training with both loss terms allows us to construct a non-trivial solution, albeit including some kind of oscillation with a decreasing amplitude over time.

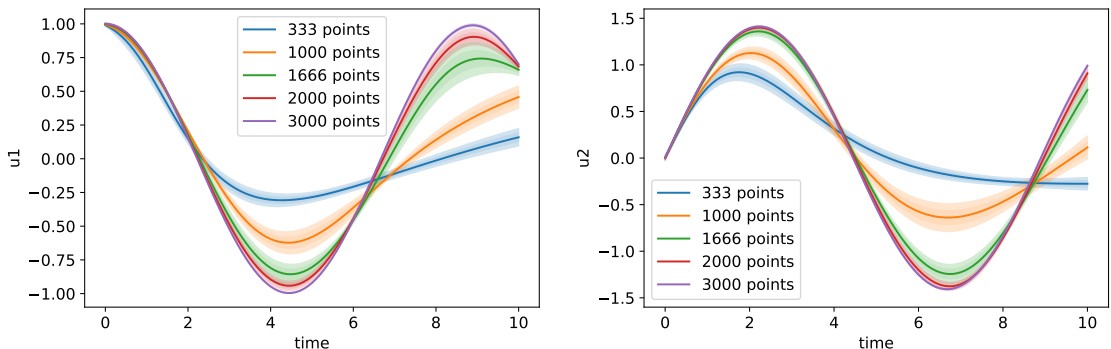

Figure 1: Learned harmonic oscillator, $u(t)$ on the left and $\dot{u}(t)$ on the right, for a varying number of uniformly distributed residual points. For the ensemble spread we train 10 independent models trained on different data points.

**Unlabeled or residual data**

As a first test, we look at the effect of the number of residual points and introduce ensembles for uncertainty estimation. Our basic architecture consists of five layers with 200 nodes per hidden layer. All our networks are written in PyTorch [28]. The training uses the ADAM optimizer [29] in a batch learning setup. For the loss, we choose equal contributions, $\beta = 1/2$. We train ten networks on 333, 1000, 1666, 2000, and 3000 uniformly distributed residual points in $t \in [0, 10]$. The means and standard deviations of this ensemble are shown in Fig. 1.

As expected, the approximation improves when the number of residual points increases. The initial condition is learned even from a small numbers of residual points, but a good prediction at later times requires more training points for the ODE loss. The reason is that for later times the network has to describe a time range rather than just a fixed vector. The trivial solution $u(t) = 0$, typically close to the network initialization, give the network the option to learn a shape which approximates the trivial solution with a decreasing amplitude at late times. For more residual points, the agreement with the true solution improves quantitatively at early times and qualitatively at late times.

We also see that the uncertainty from the network ensemble does not capture the poor agreement with the true solution. The different networks appear to be drawn to the same local minimum in the loss function even for different set of residual points.

**Labeled data**

For many physics problems, the training data can include more information than just a set of points in time. To judge the impact of this data in this section, we start with 1000 residual points $t_i$ and combine them with 6000 labeled points $(t, u, \dot{u})_i$. This additional information can then be used in the ODE loss of Eq.(2) directly. We could combine the two kinds of training data by pre-training the usual network using the labeled data points. Instead, we train the network alternatingly, where one step minimizes the MSE between the network prediction and the labels of the labeled data, and a second step minimizes the PINN loss from Eq.(2) on the residual points. We can consider training on labeled data as standard network training samples on existing data samples, while unlabeled or residual point first generate the information for the network training, in analogy to online training. In particle physics, efficient integration and sampling builds on a very similar combination of online and buffered or sample-based training [30, 31].

For the harmonic oscillator and its trivial solution it is clear that uniformly distributed labeled data points are not optimal. In Fig. 2 we show how the PINN training improves when we include labeled data in specific time windows, while the unlabeled data remains distributed uniformly.

The left panel shows that 6000 labeled data points close to the initial condition yields a significant improvement in the region of the labeled data points. Additionally, there is a small time interval where the PINNs learn a sensible extrapolation, breaking down at later times. The ensemble uncertainties do not cover any of the deviations from the true solution. In the right panel we position the labeled points at later times. Combined with the IC-loss this allows the networks to learn a good approximation over the entire time range. If we consider the initial condition as labeled data as well, this setup reduces our problem to a simple interpolation. Because the gap between the initial condition and the additional labeled points does not cover the first maximum of the oscillation, its position is captured by the PINN loss.

## 2.2 Gaussian likelihood with errors

The problem with the MSE loss in Eq.(2) is that it should be related to the probability of the network weights to reproduce the training data, $p(\theta|x_{\text{train}})$.* We usually do not have access to this probability, but we can use Bayes' theorem to replace it with a likelihood and a prior, ignoring the $\theta$-independent evidence [21]. We will show this derivation in more detail in Sec. 2.3 and assume for now that a loss should be given by the likelihood,

$$\mathcal{L} \sim -\log p(x_{\text{train}}|\theta)\,. \tag{4}$$

When we compute such a likelihood, physics observations and theory predictions come with uncertainties. Just like for a fit, these uncertainties should be part of the network training. The way to add them to the MSE loss is by noting that the MSE is the negative logarithm of a Gaussian likelihood assuming a constant uncertainty.

---

*For all ML-related arguments we follow the conventions of the Heidelberg lecture notes, Ref. [21].

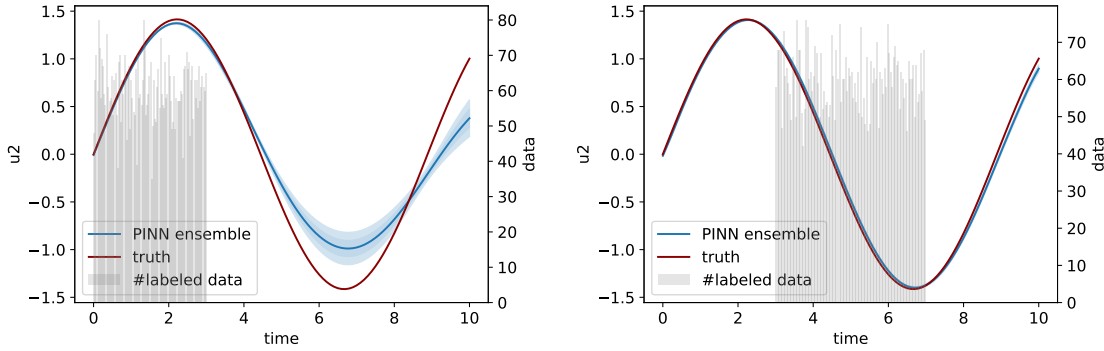

Figure 2: Learned harmonic oscillator adding labeled data points at small times (left) and intermediate times (right). The light histogram gives the distribution of training points. For the ensemble spread we train 10 independent models on different residual data points.

**Heteroscedastic loss**

This leads us directly to uncertainty quantification for PINNs via a Gaussian likelihood with a learned uncertainty function $\sigma_\theta$. For a simple, one-dimensional problem this means

$$
\begin{aligned}
\mathcal{L}_{\text{het}} &= -\log\left[\frac{1}{\sigma_\theta(x)}\exp\left(-\frac{|f_\theta(x)-f(x)|^2}{2\sigma_\theta(x)^2}\right)\right]+\cdots \\
&= \frac{|f_\theta(x)-f(x)|^2}{2\sigma_\theta(x)^2}+\log\sigma_\theta(x)+\cdots
\end{aligned}
\tag{5}
$$

During training, the network can either decrease the numerator of the MSE term or increase the denominator at the expense of increasing the normalization terms, so it learns $\sigma_\theta$ by balancing two explicit loss terms pushing the uncertainty estimate into opposite directions. This loss is referred as the heteroscedastic loss [22, 23]. The dots include the relevant prior and the irrelevant evidence and normalization constants from Bayes' theorem and the Gaussian likelihood.

Unlike Bayesian networks, the heteroscedastic loss does not distinguish between different sources of uncertainties, for instance statistical limitations and stochasticity of the training data, or a limited expressivity of the networks [21, 23, 32, 33]. Still, we will see that it is well suited to describe uncertainties from statistically limited or noisy data, in analogy to a fit maximizing a Gaussian likelihood. An open question is if it gives a reliable uncertainty estimate in the case of insufficient network architectures. On the one hand the training will focus on a, possibly, local minimum in the loss landscape, on the other hand the loss does encourage large values of $\sigma$ in case the MSE-like numerator of the log-likelihood loss cannot be further reduced.

For the PINN problem of Eq.(1), approximating a $d$-dimensional function based on $N$ residual data points, the heteroscedastic loss gives the likelihood for the network parameters to describe a solution to the differential equation,

$$
\begin{aligned}
\mathcal{L}_{\text{IC,het}} &= \frac{1}{N}\sum_{i=1}^{N}\sum_{k=1}^{d}\left[\frac{|u_{\theta,k}(t_i=0)-u_{0,k}|^2}{2\sigma_{\theta,k}(t_i=0)^2}+\log\sigma_{\theta,k}(t_i=0)\right] \\
\mathcal{L}_{\text{ODE,het}} &= \frac{1}{N}\sum_{i=1}^{N}\sum_{k=1}^{d}\left[\frac{|\dot{u}_{\theta,k}(t_i)-F_k(u_\theta(t_i))|^2}{2\sigma_{\theta,k}(t_i)^2}+\log\sigma_{\theta,k}(t_i)\right].
\end{aligned}
\tag{6}
$$

In this form, the widths $\sigma_{\theta,k}$ describe how constraining the residual points are.

In complete analogy to residual points, we can implement a heteroscedastic loss for the labeled points. In that case we start from the same regression loss as for the initial condition and introduce a learned uncertainty. This heteroscedastic uncertainty is implemented by doubling the number of output parameters of the network, half of them for the solution and half of them for the uncertainty. The training epochs exploiting residual and labeled data use slightly different losses.

We note that a more general description of the network uncertainties can be provided by Bayesian neural networks [23, 32–35]. We know that their aleatoric uncertainty, in physics terms essentially the statistical uncertainty from the training data, can be modeled using Bayesian neural networks [36]. However, for our toy example we find that Bayesian networks require significantly more training data than the heteroscedastic loss, so we skip them and instead move on to a different, new method.

## 2.3 Repulsive ensembles

An alternative way to compute the uncertainty on a network output is ensembles, provided we ensure that the uncertainty really covers the probability distribution over the space of network functions. The derivation of repulsive ensembles [21, 24] starts with the usual update rule minimizing the log-probability $p(\theta^t|x_{\text{train}})$ by gradient descent.

The update rule will be extended to an ensemble of networks, and its coverage of the network space can then be improved by a repulsive interaction in the update rule. Such an interaction should take into account the proximity of the ensemble member $\theta$ to all other members. We introduce a kernel $k(\theta, \theta_j)$ and add the interactions with all other weight configurations

$$\theta^{t+1} = \theta^t + \alpha \nabla_{\theta^t} \left[ \log p(\theta^t|x_{\text{train}}) - \sum_j k(\theta^t, \theta_j^t) \right] . \tag{7}$$

The task is to make sure that this update rule leads to ensemble members sampling the weight probability, $\theta \sim p(\theta|x_{\text{train}})$.

**Weight-space density**

To ensure this sampling property we relate the update rule, or the discretized $t$-dependence of a weight vector $w(t)$, to a time-dependent probability density $\rho(\theta, t)$. Just as in the setup of conditional flow matching networks [21], we can describe the time evolution of a system, equivalently, through an ODE or a continuity equation,

$$\frac{d\theta}{dt} = v(\theta, t) \qquad \text{or} \qquad \frac{\partial \rho(\theta, t)}{\partial t} = -\nabla_\theta \left[ v(\theta, t)\rho(\theta, t) \right] . \tag{8}$$

For a given velocity field $v(\theta, t)$ the individual paths $\theta(t)$ describe the evolving density $\rho(\theta, t)$ and the two conditions are equivalent. If we choose the velocity field as

$$v(\theta, t) = -\nabla_\theta \log \frac{\rho(\theta, t)}{\pi(\theta)} , \tag{9}$$

these two equivalent conditions read

$$\frac{d\theta}{dt} = -\nabla_\theta \log \frac{\rho(\theta, t)}{\pi(\theta)}$$
$$\frac{\partial \rho(\theta, t)}{\partial t} = -\nabla_\theta \left[ \rho(\theta, t)\nabla_\theta \log \pi(\theta) \right] + \nabla_\theta^2 \log \rho(\theta, t) . \tag{10}$$

The continuity equation becomes the Fokker-Planck equation, for which $\rho(\theta, t) \to \pi(\theta)$ is the unique stationary probability distribution.

Next, we relate the ODE in Eq.(10) to the update rule for repulsive ensembles, Eq.(7). The discretized version of the ODE is

$$\frac{\theta^{t+1} - \theta^t}{\alpha} = -\nabla_{\theta^t} \log \frac{\rho(\theta^t)}{\pi(\theta^t)} . \tag{11}$$

If we do not know the density $\rho(\theta^t)$ explicitly, we can approximate it as a superposition of kernels,

$$\rho(\theta^t) \approx \frac{1}{n} \sum_{i=1}^n k(\theta^t, \theta_i^t) \qquad \text{with} \qquad \int d\theta^t \rho(\theta^t) = 1 . \tag{12}$$

We can insert this kernel approximation into the discretized ODE and find

$$\frac{\theta^{t+1} - \theta^t}{\alpha} = \nabla_{\theta^t} \log \pi(\theta^t) - \frac{\nabla_{\theta^t} \sum_i k(\theta^t, \theta_i^t)}{\sum_i k(\theta^t, \theta_i^t)} \tag{13}$$

This form can be identified with the update rule in Eq.(7) by setting $\pi(\theta) \equiv p(\theta|x_{\text{train}})$, which means that the update rule will converge to the correct probability. Second, we add the normalization term of Eq.(13) to our original kernel in Eq.(7),

$$\nabla_{\theta^t} \sum_i k(\theta^t, \theta_i^t) \rightarrow \frac{\nabla_{\theta^t} \sum_i k(\theta^t, \theta_i^t)}{\sum_i k(\theta^t, \theta_i^t)}, \tag{14}$$

to ensure that the update rule with an appropriate kernel leads to the correct density.

**Function-space density**

So far, we consider ensembles with a repulsive force in weight space. However, we are interested in the function the network encodes and not the latent or weight representation. For instance, two networks encoding the same function could be constructed by permuting the weights of the hidden layers, unaffected by a repulsive force in weight space. This is why we prefer a repulsive force in the space of network outputs $f_\theta(x)$.

Symbolically, we can then write the update rule from Eq.(7) with the normalization of Eq.(14) as

$$\frac{f^{t+1} - f^t}{\alpha} = \nabla_{f^t} \log p(f|x_{\text{train}}) - \frac{\sum_j \nabla_{f^t} k(f, f_j)}{\sum_j k(f, f_j)}. \tag{15}$$

The network training is still defined in weight space, so we have to translate the function-space update rule into weight space using the appropriate Jacobian

$$\frac{\theta^{t+1} - \theta^t}{\alpha} = \nabla_{\theta^t} \log p(\theta^t|x_{\text{train}}) - \frac{\partial f^t}{\partial \theta^t} \frac{\sum_j \nabla_f k(f_{\theta^t}, f_{\theta_j^t})}{\sum_j k(f_{\theta^t}, f_{\theta_j^t})}. \tag{16}$$

Furthermore, we cannot evaluate the repulsive kernel in function space, so we have to evaluate the function for a finite batch of points $x$,

$$\frac{\theta^{t+1} - \theta^t}{\alpha} \approx \nabla_{\theta^t} \log p(\theta^t|x_{\text{train}}) - \frac{\sum_j \nabla_{\theta^t} k(f_{\theta^t}(x), f_{\theta_j^t}(x))}{\sum_j k(f_{\theta^t}(x), f_{\theta_j^t}(x))}. \tag{17}$$

**Loss function**

Finally, we turn the update rule in Eq.(17) into a loss function for the repulsive ensemble training. We transform the probability into a tractable likelihood loss with a Gaussian prior,

$$\log p(\theta|x_{\text{train}}) = \log p(x_{\text{train}}|\theta) - \frac{|\theta|^2}{2\sigma^2} + \text{const}. \tag{18}$$

Given a training dataset of size $N$, we evaluate the likelihood on batches of size $B$, so Eq.(15) becomes

$$\frac{\theta^{t+1} - \theta^t}{\alpha} \approx \nabla_{\theta^t} \frac{N}{B} \sum_{b=1}^B \log p(x_b|\theta) - \frac{\sum_j \nabla_{\theta^t} k(f_{\theta^t}(x), f_{\theta_j^t}(x))}{\sum_j k(f_{\theta^t}(x), f_{\theta_j^t}(x))} - \nabla_{\theta^t} \frac{|\theta|^2}{2\sigma^2}. \tag{19}$$

Here, $f_{\theta^t}(x)$ is to be understood as evaluating the function for all samples $x_1, \ldots, x_B$ in the batch.

To turn the update rule into a loss function, we flip the sign of term in the gradient, divide it by $N$ to remove the scaling with the size of the training dataset, and sum over all members of the ensemble. Since the gradients of the loss function are computed with respect to the parameters of all networks in the ensemble, we need to ensure the correct gradients of the repulsive term using a stop-gradient operation, denoted with an overline $\overline{f_{\theta_j}(x)}$. The loss function for repulsive ensembles then reads

$$
\mathcal{L} = \sum_{i=1}^{n} \left[ -\frac{1}{B} \sum_{b=1}^{B} \log p(x_b | \theta_i) + \frac{1}{N} \frac{\sum_{j=1}^{n} k(f_{\theta_i}(x), \overline{f_{\theta_j}(x)})}{\sum_{j=1}^{n} k(\overline{f_{\theta_i}(x)}, \overline{f_{\theta_j}(x)})} + \frac{|\theta_i|^2}{2N\sigma^2} \right] . \tag{20}
$$

The prior has just become an L2-regularization with prefactor $1/(2N\sigma^2)$, like in a Bayesian neural network.

**Kernel in function space**

A typical choice for the Kernel introduced in Eq.(12) is a Gaussian. For the loss in Eq.(20) this has to be a Gaussian in the multi-dimensional function space, evaluated over a sample,

$$
k(f_{\theta_i}(x), f_{\theta_j}(x)) = \prod_{b=1}^{B} \exp\left( -\frac{|f_{\theta_i}(x_b) - f_{\theta_j}(x_b)|^2}{h} \right) . \tag{21}
$$

The width $h$ should be chosen such that the width of the distribution is not overestimated while still ensuring that it is sufficiently smooth. This can be achieved with the median heuristic [37],

$$
h = \frac{\text{median}_{ij} \left( \sum_b |f_{\theta_i}(x_b) - f_{\theta_j}(x_b)|^2 \right)}{2 \log(n+1)} , \tag{22}
$$

with the number of ensemble members $n$.

## 2.4 Uncertainties

To show that we can describe uncertainties of PINNs using a heteroscedastic loss and repulsive ensembles, we use the harmonic oscillator toy model from Sec. 2.1. The only difference is that we, for instance, distribute the labeled data points such that they become sparse for late times, to see if we can track this statistic uncertainty in the training data in the uncertainty of the network output.

**Sparse and stochastic data**

First, we look at the trained network and its uncertainty estimate if we only include labeled data points and reduce the density of training data towards late times. We can do this without and with noise in the labeled data. This way, the training has no access to the late-time regime. In our setup the labels $u$ and $\dot{u}$ are separate, so this network training also misses all information about the differential equation. The decreasing distribution of labeled data points is given in the background histogram of Fig. 3, creating a smooth extrapolation problem towards late times for a simple regression.

The left panel demonstrates the effect of increasingly sparse data without noise. Indeed, the uncertainty increases with time, as the density of labeled training points decreases [38].

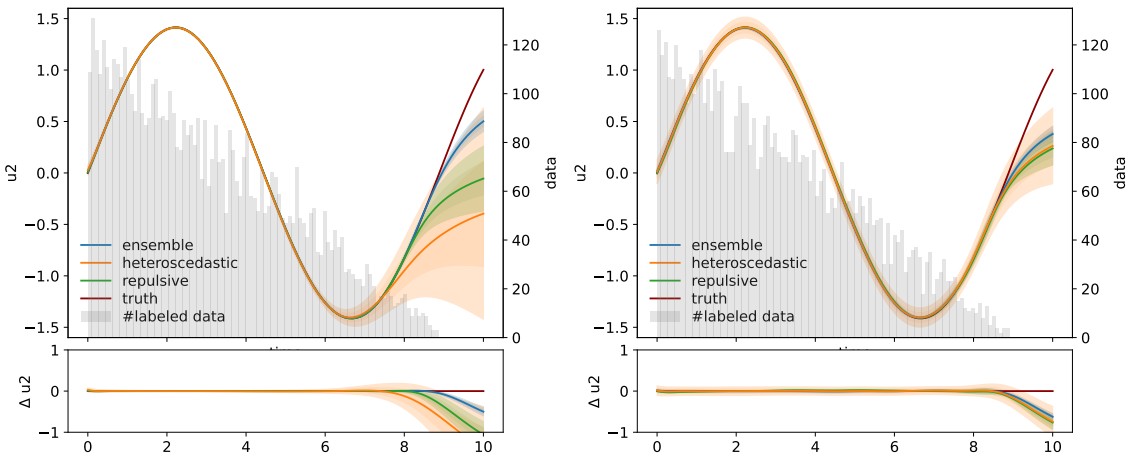

Figure 3: Learned harmonic oscillator with sparse training data at late times. For the training we only use labeled data points, defining a simple regression task. In the left panel the training data is exact, in the right panel it includes noise. The error bars correspond to 68% and 95% CL.

Both, the repulsive ensemble and the heteroscedastic network deviate from the true solution for $t > 8$, which means they have learned the shape of the minimum even though there is very little data beyond $t = 6$. The repulsive ensemble remains more stable than the heteroscedastic network, which can be explained by the stabilizing effect of ensembling. For both, the heteroscedastic network and the repulsive ensembles, the error bar increases fast enough to cover the deviation from the true solution to $t = 9$. Beyond this point the error bar is not conservative in covering the uncertainty related to missing training data altogether. The classic ensemble without repulsive term happens to guess the correct solution reasonably well, but without a meaningful spread.

In the right panel of Fig. 3 we see what happens when we switch to noisy data. Now, the labeled data points still encode the details of the differential equation, but with Gaussian noise on the $u$ and $\dot{u}$ information of mean zero and width 0.1. The heteroscedastic network captures this stochasticity as an additional source of uncertainty over the entire time range, while the members of the (repulsive) ensemble each determine the best solution without a visible spread. On the other hand, in this case it is not clear how useful the heteroscedastic uncertainty is, given that the noisy data does allow all networks to learn the true distributions very well. At late times, the noise has a counter-intuitive effect on the extrapolation; all predictions become better, and the the reduced uncertainties confirm this trend. The central values and the error bars for the heteroscedastic network and the repulsive ensembles loose all their reliability in the region without data, $t > 9$.

The simple bottom line of both tests, with and without noise, is that extrapolation for a simple regression task works as long as there is some data, and beyond this point it fails. This is true for the central value learned by the network and for the uncertainty estimate. We emphasize that in many practical applications for instance in particle physics this uncertainty estimate can still be used, because out-of-distribution data is just a limit of increasingly sparse training data.

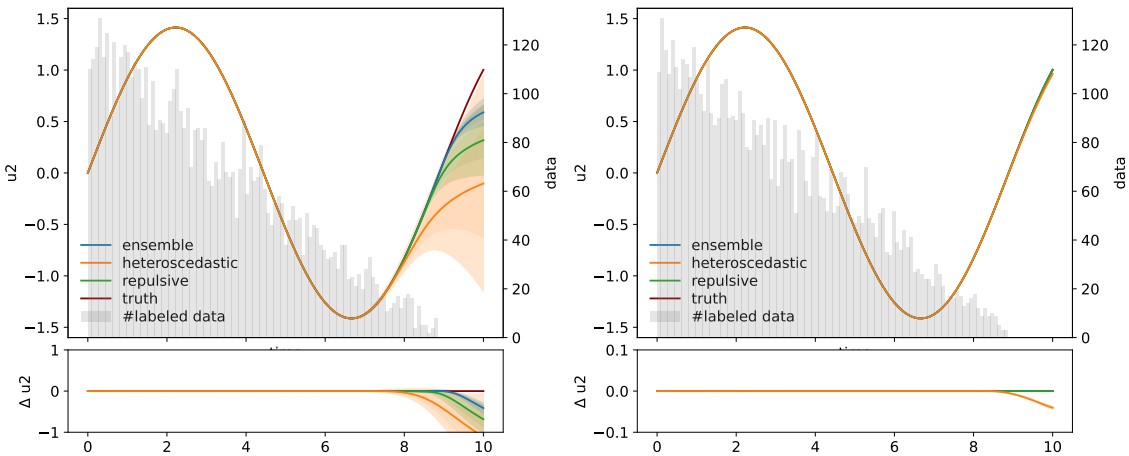

Figure 4: Learned harmonic oscillator adding the ODE loss enforcing the differential equation. For the left panel the additional residual points are distributed like the labeled point, for the right panels we add 10000 residual point uniformly over time. The error bars correspond to 68% and 95% CL.

### ODE extrapolation

The strength of PINNs is that they can extrapolate to regions without labeled data, using additional residual data trained with the ODE loss. At these points the network can confirm that its output fulfills the differential equation. We train with the two datasets alternatingly, one epoch using the labeled data point and one epoch using residual points, both computing the loss in Eq.(6).

First, we include residual data with the same time distribution as the labeled data. In practice, we strip the labeled data of the additional information and add the remaining $t$-values as residual point. In the left panel of Fig. 4 we see that the PINNs become slightly more accurate at large times than they are in Fig. 3. This is true at least in the case without noise, while we have checked that the improvement is not visible for noisy data. This means that with labeled and residual training data covering the same time range, the network does not learn the differential distributions precisely enough to provide a reliable description towards late times. The learned network uncertainty confirms the behavior of the central prediction.

Second, we add 10000 residual training points equally distributed over time. Without noise, these models reproduce the true function extremely well, over the entire time range and with correspondingly small uncertainties from the heteroscedastic loss as well as the repulsive ensembles. The uniformly distributed residual points leave no wiggle room to the network training anymore and learn the full time range. We note, however, that this does not count as an extrapolation, because the residual data also covers the entire range.

### Interpolation turning extrapolation

In a sufficiently high number of dimensions, even an apparent interpolation relies on such a low density of training data that it resembles the typical extrapolation illustrated in Fig. 3. For the extrapolation we know that both, the heteroscedastic loss and the repulsive ensembles assign an increasing error bar towards the data-deprived region, with a conservative uncertainty

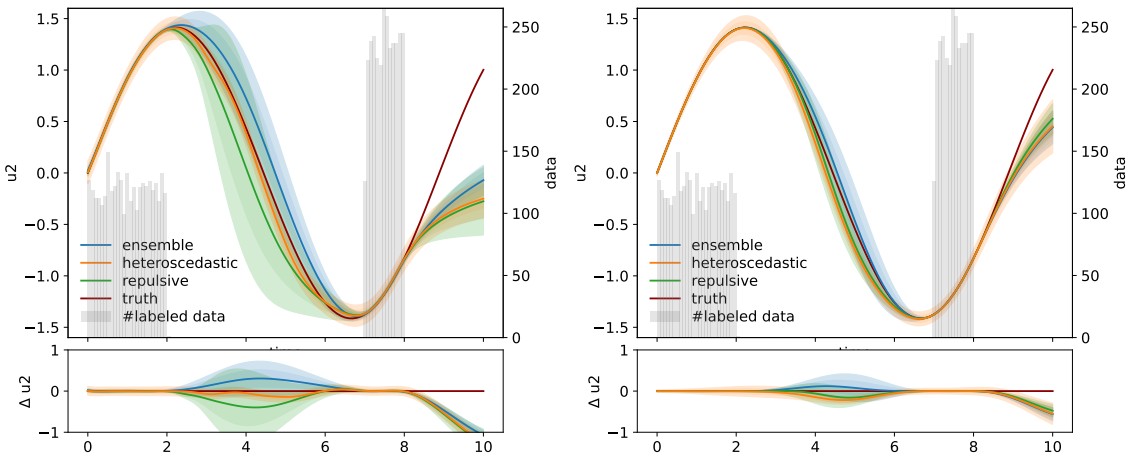

Figure 5: Learned harmonic oscillator with split training data and no noise. In the left panel we only use the labeled training point, in the right panel we add residual points distributed the same way as the labeled points. The error bars correspond to 68% and 95% CL.

estimate for as long as there is training data. The question is if the same happens for a wide interpolation.

For an extrapolation-like interpolation we assume training data at small times, $t = 0 < 2$, and $t = 7 \ldots 8$. This forces the network to interpolate over a large time window and extrapolate to very late times. In the left panel of Fig. 5 see that the wide interpolation challenges the usual ensemble of networks, indicated by the poor agreement with the true solution. The spread of the classic ensemble barely covers the difference from the truth. The situation improves with repulsive ensembles, which provide a better central prediction and much more conservative error bars in both sparse regions. Especially for the late times we see that the uncertainty assigned but the repulsive ensembles covers the deviation from the true solution well. In the interpolation region the heteroscedastic network covers a much smaller family of functions. While the central value deviates from the true solution at a similar level as the repulsive ensembles, the error bar is smaller and not really conservative for the extrapolation.

In the right panel of Fig. 5, we again add residual data following the same distribution as the labeled data. This means the network can learn the differential equation using the ODE loss. From Fig. 4 (left) we know that this has hardly any effect on regions with enough data or on actual extrapolation. However, here we see that the residual data and the ODE loss have a significant effect on the uncertainty estimate for the wide interpolation.

As a final remark — given that we know that neural networks are extremely good at interpolating, the question becomes what we expect from an error bar in the interpolation region. Either we argue that the network should consider a wide interpolation an extrapolation and admit that there is not enough data to capture possible features in the sparsely probed region. In that case the error bar should be large. Or we trust the network to interpolate well, under the assumption that there are no additional features, in which case a small uncertainty reflects the confidence of the network training.

## 3  Supernova PINNulator

The computation of the distance moduli $\mu$ of the type Ia supernovae make a compelling use case for PINNs in cosmology. For a known Hubble function the luminosity distances are computed through integration, or equivalently by solving a simple differential equation

$$\mu = 5\log_{10} d_L(z,\lambda) + 10 \quad \text{with} \quad d_L(z,\lambda) = (1+z)\,c \int_0^z \mathrm{d}z' \frac{1}{H(z',\lambda)}\,. \tag{23}$$

Depending on the assumptions on the relevant components of the universe, we can follow different strategies. In this section we focus on a two-fluid cosmology including dark matter and dark energy, $w$CDM, assuming a constant $w(z) < -1/3$ to ensure accelerated expansion. The cosmologicial constant $\Lambda$ with $w = -1$ is a particular sub-case. If we only assume the FLRW-symmetries, the Hubble function $H(z)$ can take any form allowed by the data. We come back to this second option in Sec. 4.

**Luminosity-distance PINN**

PINNs can learn to predict luminosity distances [39] and hence distance moduli, when trained to emulate the solution of the differential equation as a function of redshift. The resulting emulator, approximating the distance moduli as in Eq.(23), can be used to speed up inference.

The Hubble function depends on a set of cosmological parameters $\lambda$, and we carry this additional argument through our derivation. To apply PINNs as illustrated in Eq.(1), the luminosity distance is expressed through the ODE

$$\frac{\mathrm{d}\tilde{d}_L(z,\lambda)}{\mathrm{d}z} - \frac{\tilde{d}_L(z,\lambda)}{1+z} - \frac{1+z}{\tilde{H}(z,\lambda)} = 0 \quad \text{with} \quad d_L(0,\lambda) = 0\,. \tag{24}$$

Here, $\tilde{d}_L = d_L H_0/c$ and $\tilde{H}(z,\lambda) = H(z,\lambda)/H_0$ are de-dimensionalized and ensure solutions of order unity. This makes PINN training more stable [40]. To learn the solution to Eq.(24), we choose the cosmological parameters and the functional form for the Hubble function to conform to a flat two-fluid cosmology where the dark energy component has a constant equation of state $w$, similar Ref. [41],

$$\frac{H(z,\lambda)}{H_0} = \sqrt{\Omega_m(1+z)^3 + (1-\Omega_m)(1+z)^{3(1+w)}}\,. \tag{25}$$

Our PINN tracks three cosmological input parameters denoted as $\lambda$: (i) the redshift $z$; (ii) the energy density of matter $\Omega_m$; and iii) the dark energy equation of state parameter $w$. In this subsection, we fix the Hubble parameter to 70 km/s/Mpc.

The two relevant losses defined in Eq.(2) can be read off Eq.(24)

$$\mathcal{L}_{\text{IC}} = \frac{1}{N}\sum_{i=0}^{N}\big[d_{L,\theta}(0,\lambda_i)\big]^2$$

$$\mathcal{L}_{\text{ODE}} = \frac{1}{N}\sum_{i=0}^{N}\left[\frac{\mathrm{d}d_{L,\theta}(z_i,\lambda_i)}{\mathrm{d}z} - \frac{d_{L,\theta}(z_i,\lambda_i)}{1+z_i} - \frac{1+z_i}{H(z_i,\lambda_i)}\right]^2\,. \tag{26}$$

The index $i$ counts $N$ elements $(z,\lambda)_i$, generated uniformly over the relevant parameter ranges.

As in the toy example, we construct heteroscedastic versions of the MSE losses to learn the uncertainties from the training data,

$$\mathcal{L}_{\text{IC,het}} = \frac{1}{N} \sum_{i=0}^{N} \left[ \frac{d_{L,\theta}(0, \lambda_i)^2}{2\sigma_\theta(0, \lambda_i)^2} + \log \sigma_\theta(0, \lambda_i) \right] \tag{27}$$

$$\mathcal{L}_{\text{ODE,het}} = \frac{1}{N} \sum_{i=1}^{N} \left[ \frac{1}{2\sigma_\theta(z_i, \lambda_i)^2} \left( \frac{\mathrm{d}d_{L,\theta}(z_i, \lambda_i)}{\mathrm{d}z} - \frac{d_{L,\theta}(z_i, \lambda_i)}{1+z_i} - \frac{1+z_i}{H(z_i, \lambda_i)} \right)^2 + \log \sigma_\theta(z_i, \lambda_i) \right].$$

Our small network uses five hidden layers with 100 nodes each. The one-dimensional output approximates the luminosity distance. The $10^5$ residual training points are generated uniformly in the ranges $z \in [0, 1.8]$, $\Omega_m \in [0, 1]$, and $w \in [-1.6, -0.5]$. We will see that the network training is good enough that we do not have to consider labeled data for the PNN emulator. A similar model was used as an emulator in Ref. [39], to constrain the matter density and the equation of state parameter using the Union2.1-dataset [42–44].

**Luminosity-distance emulator**

Figure 6 demonstrates the accuracy of the PINN emulator assuming the best fit parameters of the Union2.1 dataset. For this parameter choice, the left panel shows that the heteroscedastic uncertainty on the trained PINN emulators are more than an order of magnitude smaller than the experimental uncertainties. The right panel shows that the spread of ten MSE trained PINNs is larger than the uncertainty estimation obtained when training with a heteroscedastic loss.

We can understand this behavior from the training. If we only rely on residual points, the solution is probed exactly for a given redshift. The heteroscedastic error will not be affected by stochasticity or noise, but capture the limitations from the expressivity of the neural network. In addition, the heteroscedastic training doubles the network output and allows the network to adjust the central prediction and the error as a function of time. Rather then trying to adjust a network with limited expressivity to data of arbitrary precision, it can offload some problems to the learned uncertainty, which can, and does, stabilize the training and the ultimate agreement with the true solution.

To allow for cosmological inference, an emulator of distance moduli also has to approximate the solution to the differential equation away from the best fit parameters. To test the

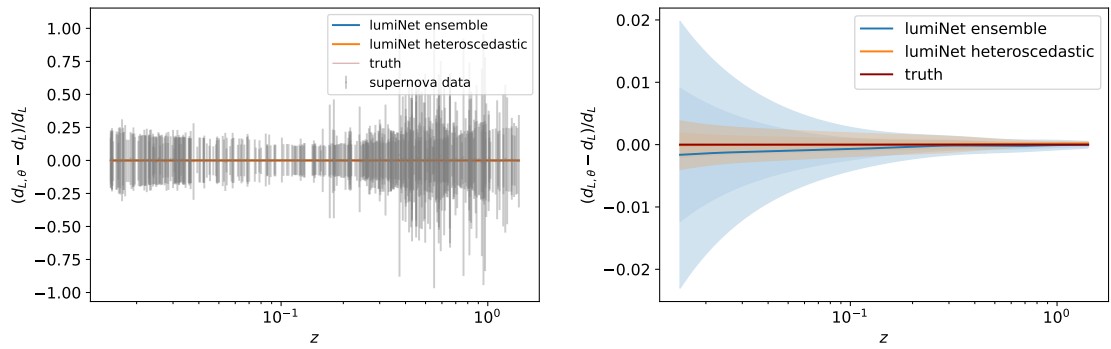

Figure 6: Learned luminosity distance from residual points only. The left panel compares the heteroscedastic PINN uncertainty to the experimental uncertainties in the Union2.1 dataset. The right panel shows the relative difference between the learned and true solutions. For the ensemble spread we train 10 independent models on different data points.

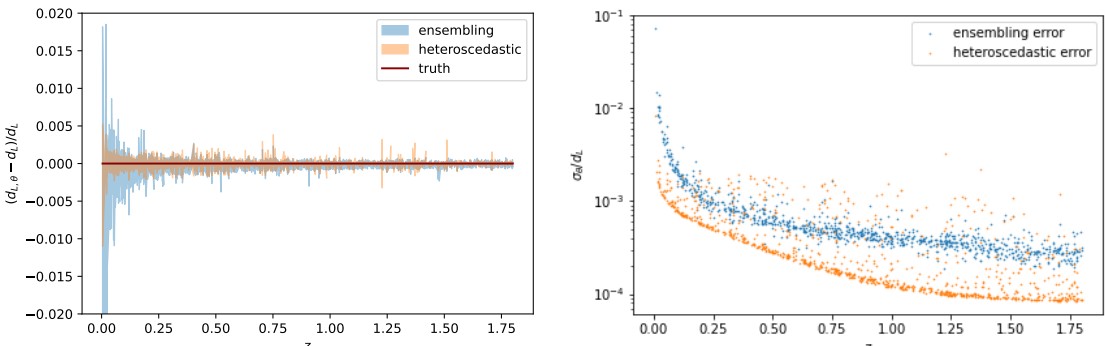

Figure 7: PINN accuracy for data points uniformly sampled from the same cosmological parameter ranges as the training points. The left panel shows the error bands around the true solution, the right panel the evolution of the ensemble spread and the heteroscedastic uncertainty with redshift.

reliability of the PINN we generate 1000 test data points from the same distribution as the training data. For this test data we first compute the true luminosity distance using Eq.(24). Then, we generate the learned luminosity distances and their uncertainties from the PINN. The left panel of Fig. 7 shows the deviation of the PINN prediction from the true solution. The spread of the ensemble trained with an MSE loss deviates from the truth by less than two percent. The heteroscedastic training improves this agreement to better than one percent. However, in the right panel of Fig. 7, we also see that the relative uncertainties grow rapidly for small redshifts, because the initial condition for the luminosity distance is also small. This requires higher absolute precision at small redshifts.

Nevertheless, we find that especially the PINNs trained with the heteroscedastic loss are extremely precise even without resorting to labeled data training. This is definitely sufficient to be used as an emulator for the luminosity distance for the Union2.1 or Pantheon+ [45] data, which come with experimental errors of around 10%.

## 4 Supernova PINNference

The previous section demonstrates that PINNs can learn and emulate luminosity distances arising for a given parameterized Hubble function as a solution to a differential equation. Inference inverts this process. Now the errors on a dataset need to be mapped onto a corresponding uncertainty on the inputs, either discrete parameters or a neural network-represented free Hubble function, as we will do next.

The formulation of a differential equation with a free function $f_\phi(t) \approx f(t)$ to be represented by a neural network, similar to [46], expands the structure of Eq.(1) to

$$\dot{u}(t) = F(u(t), t, f(t)) \quad \text{with} \quad u(0) = u_0 \ . \tag{28}$$

We extract information on the differential equation including $f(t)$ by training a network $u_\theta(t)$ on the labeled data. This network should fulfill the differential equation with the true function $f(t)$. This function is approximated with a second network $f_\phi(t)$. Given $N$ labeled data points

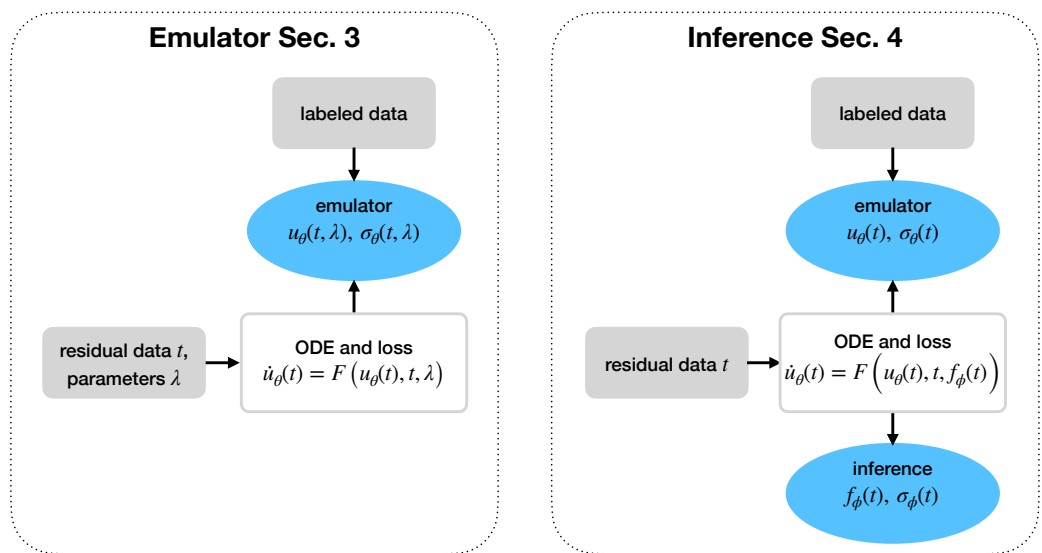

Figure 8: Illustration of the PINN emulation and inference setups.

$(t, u)_i$ and $M$ residual points $\tilde{t}_j$ the training uses the loss functions

$$
\begin{aligned}
\mathcal{L}_{\text{Data}} &= \frac{1}{N} \sum_{i=1}^{N} [u_\theta(t_i) - u_i]^2 \\
\mathcal{L}_{\text{ODE}} &= \frac{1}{M} \sum_{j=1}^{M} \left[ \dot{u}_\theta(\tilde{t}_j) - F(u_\theta(\tilde{t}_j), \tilde{t}_j, f_\phi(\tilde{t}_j)) \right]^2 .
\end{aligned} \tag{29}
$$

The data loss plays the same role as $\mathcal{L}_{\text{IC}}$ in Eq.(2). This ensures $f_\phi(t) \approx f(t)$ for all times covered by the residual points, as long as $u_\theta$ is sufficiently accurate. The information on $f(t)$ is first extracted from the data using the network approximating the differential equation via $u_\theta$. In a second step, the differential equation is used to infer the function itself. In all numerical experiments the losses are combined by alternating between epochs using only one of them. The network structure and training are illustrated in Fig. 8.

Figure 9 tests this setup for a cosmological model defined by Eq.(25) with $w$ fixed to the best-fit value of the Union2.1 dataset. This reconstruction uses a dense network with five hidden layers with a width of 100 nodes for both networks encoding the luminosity distance $d_\theta$ and the Hubble function, $H_\phi$. The reconstruction is performed using $10^4$ residual points and $10^3$ labeled data points, the same order of magnitude as current surveys [45, 47]. For each epoch of training with the ODE loss ten epochs are trained with the data loss. Alternating between epochs of training on the ODE and training on data allows to control the relative weight between fulfilling the ODE and fitting data. This reconstruction of the (inverse) Hubble function is performed without any input on the particular model used to generate the data. On this synthetic data set the Hubble function can be learned almost perfectly.

## 4.1 Uncertainty estimation

A key part of ML-inference is the control over uncertainties affecting the network training. As demonstrated in Sec. 2.3, a repulsive ensemble of networks extracts a meaningful uncertainty, especially in regions with sparse data. To confirm this error estimate, we also use a

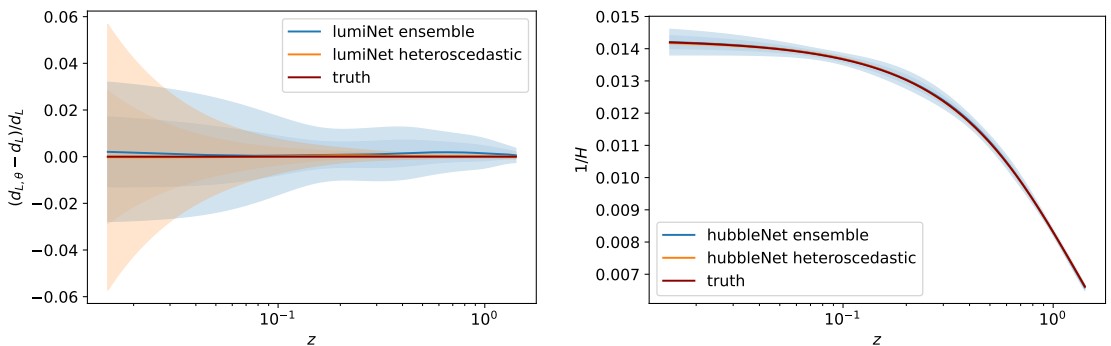

Figure 9: Reconstruction of the Hubble function using PINNs with an MSE loss. The left plot depicts the luminosity distance approximation compared to the true value with an ensembling error bar derived from ten models. The right-hand side depicts the corresponding Hubble reconstruction.

heteroscedastic loss, in particular when there are no significant gaps in the data.

Combining the learned luminosity distance $\tilde{d}_{L,\theta}$, with uncertainty $\sigma_\theta$, and the ODE in Eq.(24), every luminosity distance value contributes to the reconstruction of the Hubble function as

$$\frac{1+z_i}{\tilde{H}(z_i)} \approx \frac{d\tilde{d}_{L,\theta}(z_i)}{dz} - \frac{\tilde{d}_{L,\theta}(z_i)}{1+z_i}. \tag{30}$$

To include $\sigma_\theta$, both $\tilde{d}_{L,\theta}(z_i)$ and $d\tilde{d}_{L,\theta}(z_i)/dz$ need to be drawn from their respective probability distributions. By using a heteroscdastic loss the luminosity distance at each redshift is assumed to follow a normal distribution $\mathcal{N}(\tilde{d}_{L,\theta}(z_i), \sigma_\theta^2)$. Since samples of the luminosity distance are generated using a standard Gaussian, the width of the derivative distribution is $d\sigma_\theta/dz$. Generating samples from these distributions and inserting them into Eq.(30), it is possible to generate a distribution of of Hubble function values for each redshift.

A second network can then learn $\tilde{H}_\phi$ with an uncertainty $\sigma_\phi$ based on the luminosity distance network $d_{L,\theta}$, where both networks are trained to fulfill Eq.(30) and fit the data. The uncertainty on $\tilde{H}_\phi$ is learned using the heteroscedastic loss of Eq.(27). This uncertainty can be interpreted as the uncertainty on $(1+z)/\tilde{H}_\phi(z)$ under the assumption that for each redshift $d_{L,\theta}$ fulfills the differential equation correctly. This allows us to reduce the loss function to the expression

$$\mathcal{L}_{\text{Hubble, het}} = \frac{1}{N}\sum_{i=1}^{N}\left[\frac{\left(\frac{1+z_i}{\tilde{H}(z_i)} - \frac{1+z_i}{\tilde{H}_\phi(z_i)}\right)^2}{2\left(\sigma_\phi(z_i)\right)^2} + \log\sigma_\phi(z_i)\right]. \tag{31}$$

The Hubble function is then approximated by a normal distribution in $(1+z_i)/\tilde{H}_\phi(z_i)$ with variance $\sigma_\phi^2(z_i)$.

The combination of Eqs.(30) and (31) allows us to optimize $H_\phi$ and $d_{L,\theta}$ simultaneously. The mean value and uncertainty of $d_{L,\theta}$ appear in the sampling of $d_L(z_i)$, allowing the network parameters $\theta$ to influence the loss. For every epoch trained using the differential equation loss, the PINN for the luminosity distance is also trained to match the labeled data. The ratio of labeled data epochs to ODE epochs is a training hyper-parameter.

## 4.2 Noisy data

To analyse real data we have to allow for noise in solving the inverse problem. We consider two datasets, Union2.1 [42–44] and Pantheon+ [45]. To use them as labeled training data, we convert them into luminosity distances with corresponding error bars using Eq.(23). Assuming that the data follows a multivariate normal distribution, we can generate a set of luminosity distances per redshift using the mean and the covariance matrix from the actual data.

The resulting luminosity distances and the distribution of redshifts for the ensemble of synthetic datasets is depicted in Fig. 10. Typical errors are around 10%, and the data becomes sparse towards large redshift. The newer Pantheon+ dataset covers a larger range in redshifts and includes three times as many supernovae.

In this section the luminosity distance is learned as $d_\theta$, using five layers with 100 nodes each. This network is trained on the labeled data. The inverse Hubble function is modeled with a second network with five layers and 200 nodes wide. As suggested in Ref. [40] we impose the boundary condition of the luminosity distance network by learning $(d_L/z)_\theta$ and multiplying by $z$ later. In addition, random Fourier features [48] significantly reduces the required training time. For each dataset, the training data for each epoch is generated from the luminosity distance distribution shown in Fig. 10. The resulting ensemble of luminosity distances scatters around the mean at each redshift, which can be captured by the heteroscedastic loss.

In Fig. 11 we show the reconstruction of the Hubble function from both datasets. We shown the learned luminosity distance and the the reconstructed Hubble function, comparing a heteroscedastic network, an ensemble of MSE networks and a repulsive ensemble. Similar to Sec. 2.4, the ensemble and the repulsive ensemble using the labeled data region does not capture the noise, whereas the heteroscedastic uncertainty of the luminosity distance does. The reconstructed Hubble function is consistent with a $w$CDM approximation of the Hubble function from a direct fit of a parameterized model.

The sharp feature in the Hubble function reconstruction from the Union2.1 dataset can be understood from Eq.(30). The uncertainty of the Hubble function is approximately the quadratic mean of the uncertainty of the derivative of the luminosity distance and the uncer-

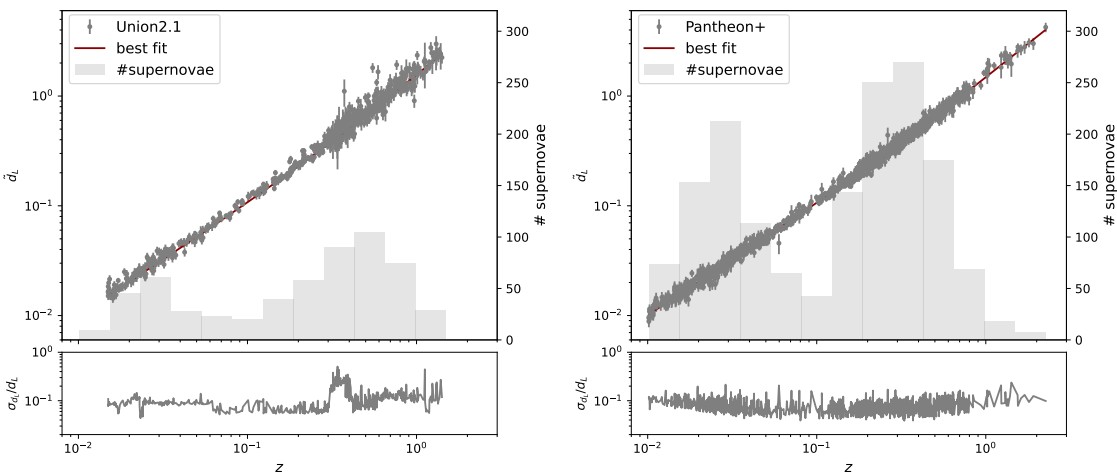

Figure 10: Generated redshift dependencies of the luminosity distance values of the Union2.1 (left) and Pantheon+ data (right). The histograms capture the distribution of the supernovae in redshift. The lower sub-panels show the relative error bars on the luminosity distances.

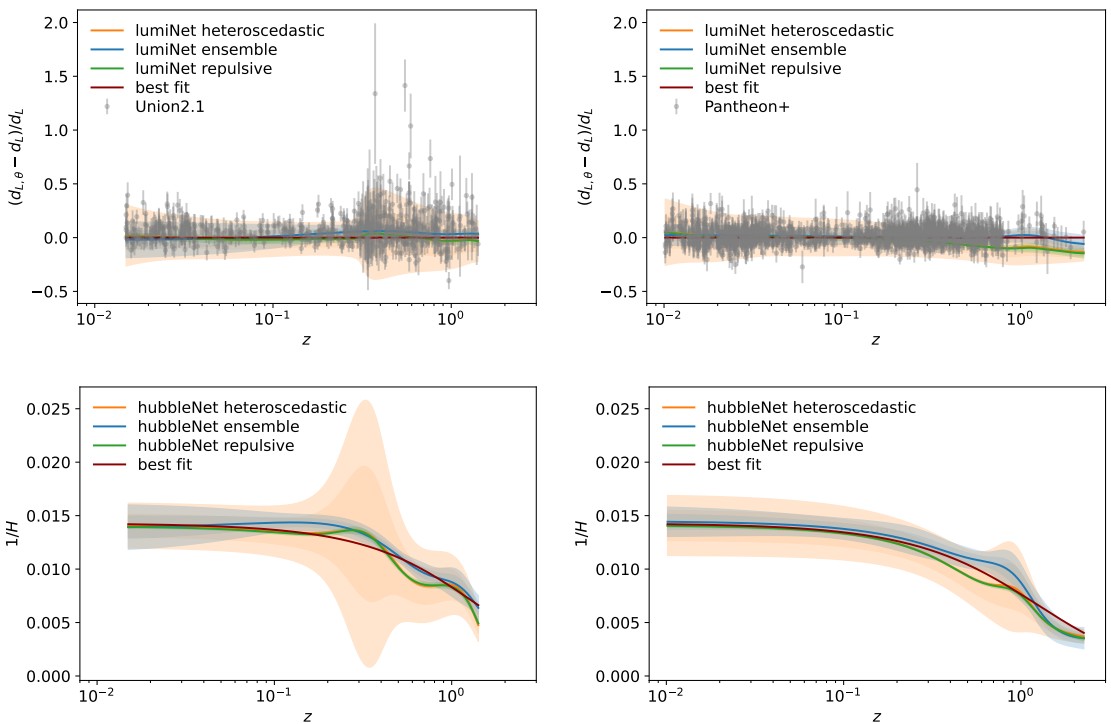

Figure 11: Top: PINN-learned luminosity distance from the labeled data, derived from the Union2.1 (left) and Pantheon+ (right) data. Bottom: learned inverse Hubble function from the two datasets.

tainty of the luminosity distance itself. Fast changes in the width and scatter of the labeled data points with redshift, see Fig. 10, leverage fast changes in the predicted error bars of the Hubble function. The sharp increase in the uncertainty of the reconstructed Hubble function at redshift 0.3 corresponds to the change in the uncertainty in the luminosity distance leading to a maximum in the uncertainty.

The reconstruction of the Hubble function in Eq.(30) relies on the assumption that the network approximating the luminosity distances fulfills the differential equation exactly. The deviation from the a true solution can be approximated by inserting both networks into the differential equation. When applying PINN inference to our datasets, this deviation is small compared to the predicted uncertainties from the spread of the data.

## 4.3 Dark energy equation of state

Finally, we convert the inferred, parameter-free Hubble function $H(a)/H_0$ to an equation of state function $w(a)$. Using the general relation [49],

$$\frac{H^2(a)}{H_0^2} = \frac{\Omega_m}{a^3} + (1 - \Omega_m) \exp\left[-3 \int_1^a da' \frac{1 + w(a')}{a'}\right], \tag{32}$$

we extract $w(a)$ by differentiation,

$$w(a) = -\frac{1}{3} \frac{\mathrm{d}}{\mathrm{d}\log a} \log\left[\frac{H^2(a)}{H_0^2} - \frac{\Omega_m}{a^3}\right] - 1. \tag{33}$$

We use $\Omega_m = 0.28$, as suggested by the Union2.1 dataset. Naturally, this differentiation introduces a larger uncertainty when transitioning from the inferred Hubble function $H$ to the

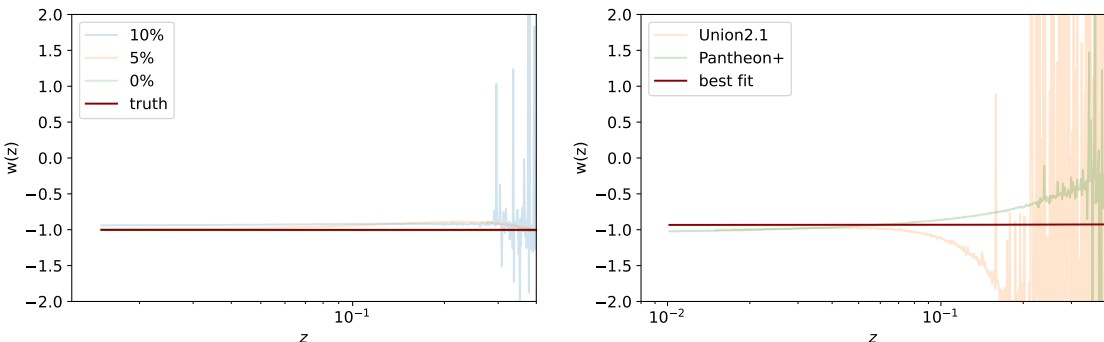

Figure 12: Inferred dark energy equation of state. The left panel uses simulated date with increasing assumed error bars. The right panel uses the Union2.1 and the Pantheon+ dataset, propagating the error bars estimated by the collaborations through the PINN-inference.

equation of state $w$.

Performing a test on $10^3$ simulated supernovae, uniformly in redshift, leads to the left panel of Fig. 12. It demonstrates that we can reconstruct $w(z)$ with small uncertainties. Increasing the observational uncertainty to 5% or 10% shows a commensurate effect on uncertainty of the inferred $w(z)$. In terms of redshift, the uncertainty becomes large beyond $z \simeq 0.3$ for realistic errors, which is partially caused by the uncertainty of the PINN far away from its initial conditions. But more importantly, dark energy has a small influence on the Hubble function at high redshift, rendering $w(z)$ effectively unconstrained. Technically, by approaching $H(a)^2 \simeq \Omega_m/a^3$ at sufficiently high redshifts leads to a diverging logarithmic derivative in Eq.(33).

In the right panel of Fig. 12 we show the reconstruction of $w(z)$ from our two datasets. The matter density for each of dataset is again assumed to be the best fit value. At small redshifts our inference method constrains $w(z)$ well, but the uncertainties of the labeled data do not leave any sensitivity beyond $z \gtrsim 0.3$.

## 5   PINNclusions

Physics-informed neural networks are trained on the output of a parameterized system of differential equations. They can predict solutions for given parameters with a proper interpolation between parameter choices. This emulation of the space of ODE solutions provides tremendous speed-ups and therefore an excellent tool for statistical inference. The focus of our investigation was the error-awareness or uncertainty estimation of PINNs. For this purpose we have compared a heteroscedastic loss and repulsive ensembles, confirming that PINNs extrapolate into regions of sparse or low-quality data, while sensibly increasing their learned error in these regions. Testing these aspects with the harmonic oscillator as a toy example confirms the fundamental behavior of PINNs.

The functionality of PINNs as emulators was then verified with luminosity distances as functions of redshift for a conventional dark-energy dominated Friedmann-Robertson-Walker universe. PINNs correctly predict the luminosity distance for a given redshift over a wide range of dark energy equation of state parameters, without solving a differential equation, or equivalently in this case, performing a numerical integration.

Using PINNs for inference rather than emulation requires a statistical inversion, i.e. a map-

ping of the experimental uncertainty back to the parameterization. Applied to the supernova example, PINNs allow for an uncertainty-aware reconstruction of the Hubble function without any predefined parameterization. The Hubble function is reconstructed by the PINN including an error estimate. They discover peculiarities in the data, such as the sudden increase in error in the Union-data set at $z \simeq 0.3$, reflecting a large uncertainty in the reconstructed Hubble function. Re-expressing the Hubble-function with the dark energy equation of state function derived for a fixed matter density shows weaker constraints, as the increase in error is driven by the derivative transitioning from $H(a)$ to $w(a)$.

## Acknowledgements

We would like to thank Manuel Haussmann for recommending repulsive ensembles to us, Theo Heimel for his advice and his implementation of the repulsive ensmbles, Ullrich Köthe for valuable technical help with PINNs, and Benedikt Schosser for help with the Pantheon+ dataset. This work was supported by the Deutsche Forschungsgemeinschaft (DFG, German Research Foundation) under Germany's Excellence Strategy EXC 2181/1 - 390900948 (the Heidelberg STRUCTURES Excellence Cluster).

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
