# Peer review of "PINNferring the Hubble Function with Uncertainties"

_SciPost Physics_

## Round 1 · Referee Report · Anonymous (Referee 1) · 2024-8-1

Strengths

  1. The article provides an exploration of various PINN-based methods for non-parametric inference and uncertainty estimation in the context of cosmology.
  2. The specific application to type Ia Supernova cosmology, and the adopted repulsive ensemble method with density estimation in function space are new and relevant for the community.
  3. It is useful to see different methods of uncertainty estimation in direct comparison on the same dataset.

Weaknesses

  1. The connection of the proposed methods with principled Bayesian (or Frequentist) inference remains unclear. The PINN-based results seem to be mostly of indicative value, rather than providing solutions to well-defined statistical problems (with known likelihood functions, priors, etc).
  2. The paper would greatly benefit from a direct comparison between PINN-based results and results obtained with more established techniques, making the same or comparable modelling assumptions. An example would be here to use the proposed methods to perform 2-dim parameter fits (Omega_m, w) with MCMC to mock SN Ia data (based on the d_L emulator).
  3. The paper covers a lot of ground. In particular section 2 could benefit from summarising discussion of the main results and points.

Report

In the article "PINNferring the Hubble Function with Uncertainties", the authors demonstrate how physics-informed neural networks (PINNs) can be used to emulate the luminosity-redshift distance relation and how to use them in an inference context, while exploring various strategies for uncertainty estimates. The paper provides interesting new content at the intersection of PINNs, inference and cosmology. However, I would like to see more discussion about how the proposed uncertainty measures and inference techniques are related to established (PINN-free) techniques before I can recommend publication.

Requested changes

Section 1 1. The authors seem to imply that the calculation of luminosity distances for free-form Hubble functions is a computational bottleneck in existing inference pipelines. Please clarify this point. 2. Connected with the previous point, it remains somewhat unclear what exact use-cases the authors have in mind for their proposed method. What is the base-line performance of traditional methods inference methods they aim to compare with? Please clarify.

Section 2 1. Eq. (2): Is L_ODE averaged over time? Please clarify at the level of the equation (it becomes clear later, but is confusing at this point). 2. Fig,. 1: I am surprised that the results depend so much on the number of training points, since for this very simple target additional training points do not really contribute new information about the shape of the function. Could this have something to do with rebalancing the relative contributions of L_IC and L_ODE in Eq. (2)? Please clarify this, also in connection with the previous point. 3. At the beginning of section 2.2, the authors write p(theta | x_train). For this to have a meaning, one needs a prior p(theta) and some likelihood p(x_train | theta). What is that likelihood function concretely? Or should I understand "p(theta | x_train)" to only have an indicative meaning? Please clarify. 4. Eq. (4): Should there be also evidence and prior in that equation, or colons like in Eq. (5)? Please clarify. 5. Eq. (7): What does the sum run over? 6. Eq. (8): Are the v(theta, t) in both equations the same? If yes, please use the same font. 7. "Weight-space density": Please make clear where these arguments were made first. 8. Eq. (16): I am not convinced of the step (15) --> (16), it seems ad hoc rather than an actual mathematical derivation. For instance, equation (15) depends implicitly on a prior in function space (which the authors probably want), while equation (16) depends implicitly on a prior in weight space (which they end up using). I don't see how p(f | x_train) can just replaced by p(theta | x_train). Please clarify how rigorous this step really is, and if/where these arguments have been made before. 9. Eq. (19): I would suggest to make already at the level of the equation clear that in the second term the kernels are evaluated for all x_i from the batch simultaneously. Maybe also just forward reference Eq. (21), where this is made clearer (same for Eq. (17)). 10. "Sparse and stochastic data", second paragraph: The classic ensemble method does pretty well in Fig. 3, but the authors say "without a meaningful spread". What is the definition of "meaningful spread"? 11. "Sparse and stochastic data", third paragraph: Why Gaussian noise with width (standard deviation?) 0.1? How would other values affect the results and conclusions? 12. "ODE extrapolation": The authors start this subsection saying that PINNs can extrapolate to regions without labeled data, using residual data. They end with saying that using residual data does not count as extrapolation. How does that fit together? Please clarify. 13. End of section 2: The authors discussed various methods for uncertainty quantification for PINNs, but the conclusions and usage recommendations remain unclear. I suggest to add a brief conclusions section where the results and findings are summarised for usage in Section 3 and 4, maybe with a table.

Section 3 1. Eq. (25) and afterwards: Does lambda include z or not? The list after (25) suggest it is 3-dim and includes, z, Omega_m and w, but then the authors use (z, lambda) in their notation, suggesting that lambda just contains Omega_m and w. 2. Eq. (27): Why exactly are the authors using here heterscedastic versions of the MSE losses in this case? Is this motivated from the findings in Section 2? 3. "Luminosity-distance emulator", second paragraph: The authors say that heterscedastic errors capture the limitations of the neural network expressivity only. Please support that statement with references or with Section 2 results.

Section 4 1. "...repulsive ensemble of networks extracts a meaningful uncertainty...": How does that uncertainty compare with more traditional (Frequentist/Bayesian) uncertainty estimates? Please clarify. 2. Section 4.3: The authors extract from their free-form Hubble function the equation of state. Is there any advantage of this over just parametrising the equation of state as a neural network and taking it from there? Please clarify. 3. The results should be compared - at least in some cases - with results from traditional techniques where possible. A meaningful quantitative comparison with established techniques, at least in edge cases, is in my opinion a necessary ingredient for a paper proposing a new analysis methods. I leave the details up to the authors, but one option would be to perform a parameter fit for Omega_m and w using the d_L emulator, using nested sampling or MCMC. Another one would be to use a spline as Hubble function and fit the spline coefficients.

General 1. Although I find the title and section headings very PINNtertaining (no PINN intended) I suggest to remove all PINN-puns from the section titles. 2. In general, it is not clear to me how to interpret the inference method. Is it in any sense Frequentist? Or is it Bayesian? In a Bayesian context, there necessarily is a prior for the Hubble function, and is a likelihood function connecting the measured luminosity distances with the theoretical predictions. But: (A) What is the prior for the Hubble function in the proposed method? It seems to be just implicitly defined by the flexibility of the network. (B) Is there any reason to believe that the PINN-based solution of the inference problem exactly accounts for the uncertainties of the data generation process? 3. Both of the above points would be very explicit and clear when using likelihood-based variational inference techniques (based on neural ODEs) or using simulation-based inference techniques (maybe with Gaussian process priors for the Hubble function). Are there any advantages of using PINN based methods in this context? Please clarify.

Recommendation

Ask for major revision

---

## Round 1 · Referee Report · Anonymous (Referee 2) · 2024-8-31

Strengths

  1. The authors compare multiple methods / cases, gradually building up from a simple / baseline to a more sophisticated demonstration (e.g. simple PINN -> heteroscedastic / ensemble -> repulsive; harmonic oscillator -> Hubble distance -> w(z)).
  2. The paper includes demonstrations on synthetic, as well as real observational data.
  3. Figures are generally clear and well-referenced in the text.

Weaknesses

  1. It is unclear to me (if not also to the authors) what uncertainty their method "learns": is it uncertainty in the data, in the recovered solution due to imperfect training, or the fundamental (Bayesian) inference uncertainty.
  2. A clear statement of how the present work advances the field is generally lacking (fragments in the PINNclusions). E.g. PINN emulators for SNae Ia were already used (by the same lead author) in [39]. If the emphasis is on "speeding up ODE solutions", a speed comparison is missing.
  3. The application to real data is very rushed, while scientific conclusions are not drawn at all, despite hint of mismatch with previous work ("best fit").
  4. Language is less formal than I'd expect of a scientific publication, and that detracts from the clarity of presentation. Also, symbols and important concepts (which quantity is a NN in equations and what are its input, for example; which quantities are considered to contain the "noise") are often not defined. Lastly, it is unclear why/how PINNs are informed by physics.
  5. Section 2 is extremely long and overly detailed. Some of the derivations might be better off in an APINNdix ;)

Report

The paper presents two methods for quantifying "uncertainty" in neural-network reconstructions of the Hubble parameter vs redshift. Specifically, it focuses on so-called physics-informed neural networks, which are trained to approximate both the value of a function and its derivative. Ultimately, the authors derive a non-parametric reconstruction from real observational SN Ia data (the Union2.1 and Pantheon+ compilations).
Throughout the work, the concept of "uncertainty" is extremely vague and at times used for different concepts (noise in the data, reconstruction inaccuracy, inference uncertainty). Before publication (and before application to real data; indeed, uncertainties are very superficially handled in 4.2 and not really present in 4.3), the authors must examine in detail the _validity_ of their "uncertainty" estimates, preferably in relation to a well-established statistical framework. I suggest a direct comparison with the leading non-parametric function reconstruction method: Gaussian processes, which achieve exactly what (I feel) the authors have set about with much more rigour.

Requested changes

Major comments: - The authors should understand and clearly explain/highlight which uncertainty their heteroscedastic $\sigma(t)$ models: the noise in individual data, or the uncertainty of the reconstruction; then perform "PINNference" accordingly (the former case is a likelihood, the latter a posterior) - Can the heteroscedastic approach handle uncertainties in the independent variable, i.e. in the redshift for the application? Future data sets will contain almost exclusively uncertain redshifts with very non-Gaussian posteriors from previous analyses. - Similarly, how does the method handle correlated noise, which is essential in SN Ia studies. - Section 3: the authors present an "emulator for the luminosity distance". But there are already numerous packages that calculate it, e.g. using known analytic expressions for certain models, or using conventional odeint, which supposedly the authors used to generate their training data. This begs the question, how does the emulator from section 3 advance the field? The authors should try to highlight its advantages (if any) versus other (conventional and simpler to comprehend) methods. And to anticipate a response: I do not believe the uncertainty quantification is advantageous particularly since an emulator should strive for perfection, especially if a perfect "emulator"(odeint) already exists.

Detailed comments:

  1. PINNtroduction:
  2. it seems the authors use interchangeably the notion of a theoretical model for the Hubble function and data-based reconstruction/inference; e.g. "Our approach relies only on the symmetry principles for spacetime and derives the Hubble function H(a) free of any parameterization directly from data."
  3. " our inference works great..." needs to be re-phrased to carry a concrete meaning, e.g. with a specific quoted precision.

  4. PINNcertainties:

  5. in eq. (2), \theta is not defined. It turns out only in 2.2 that this is the network weights, which needs to be mentioned earlier. In fact, throughout all of 2. it was not made clear what are the inputs and outputs of the neural network.
  6. "labelled data": the loss in eq. (2) is a supervised loss, so it also requires "labelled" data to train with. As far as I understand (which is not explained at all, and I'd strongly suggest the authors spend significant effort explaining this), the "unlabelled" training data consists of pairs $[t_i, F(u_{\theta}(t_i), t_i)]$ (the latter evaluated "on the fly" and compared with the gradient of the network with respect to $t$, which is, supposedly, an input to the network). At this point, an explicit mention that the NN is trained to approximate the gradient of the sought function is warranted. Now, the additional "labelled" data the authors add further down in the section is, in spirit, the same as the initial condition $u_0 \equiv u(0)$ used for the $\mathcal{L}_{IC}$. Hence, I'd recommend simply adding the "label"-related loss in eq. (2) from the beginning: $[u_\theta(t_i) - u(t_i)]^2$. (In fact, the authors already hint at this in the last sentence of 2.1). If, of course, I have understood correctly...
  7. 2.2 "The problem with the MSE loss in Eq.(2) is that it should be related to the probability of the network weights to reproduce the training data, $p(\theta|x_{train})$." The meaning of this sentence is extremely unclear to me. In fact, to me, it contradicts the meaning of the following paragraphs describing the heteroscedastic loss, which concerns a probability for the training data given the NN as a model. In fact, $p(\theta|x_{train})$ is not used by the authors' PINN but rather by Bayesian neural networks, which the authors explicitly contrast in the last paragraph of 2.2. Re: that paragraph, I point out that, unlike the authors' heteroscedastic loss, which represents a modification to the model, in a Bayesian sense, BNNs simply return uncertainty on the parameters under the non-modified model and thus handle a different kind of uncertainty altogether.
  8. 2.3.
  9. in the first paragraph, I assume the authors mean "maximising" the log-probability
  10. Function-space density: "we are interested in the function the network encodes and not the latent or weight representation. " summarises the confusion between uncertainty, aka scatter, as part of the model (which the authors include by predicting $\sigma(t)$) and uncertainty in the inferred parameters, which is addressed through the repulsive ensemble.
  11. Loss function (eq. (18)): the "Gaussian prior" is known as a weight-decay loss and while it is somewhat arbitrary, it has been used extensively in deep learning: it's actually implemented directly in Pytorch's Adam optimiser, which the authors use. the authors should remark on that and/or refer to previous studies -2.4.
  12. fig. 3, the labels "repulsive" and "ensemble" as two separate experiments confused me, maybe specify "non-repulsive" or similar. Also, it's curious to investigate what the NN returns... without training, i.e. optimised without training data (because the loss contains other not-data-related terms). This would correspond to the "prior". and be indicative of the "high-t" regime in fig. 3
  13. second par on p.10: reminded me that the "labels" include the quantity and its derivative with time. It would be good if the authors highlighted this explicitly in the initial presentation of the loss for labelled data and, importantly, explicitly contrast it (and discuss performance differences) with the $\mathcal{L}_{ODE}$ from eq. (2); I surmise the difference is that one uses $F(u_\theta(t), t)$, whereas the other $\dot{u}(t) \equiv F(u(t), t)$.
  14. "it is not clear how useful the heteroscedastic uncertainty is": indeed: the mean reconstruction is very close to the truth in fig. 3 (right) in comparison with the predicted noise (spread in orange); I venture the explanation that the heteroscedastic "$\sigma(t)$" does not model the posterior uncertainty but rather the noise it each individual point!! That is, the mean reconstruction is scattered around the truth by $\sigma(t)/\sqrt{N}$, where N is the number of points that contribute to the inference of $\mu(t)$ at a particular t, which is hard to quantify. A piece of supporting evidence: the reconstruccted $\sigma(t)$ is practically constant in fig. 3 (right) because the injected noise is, unlike the reconstruction uncertainty, which should increase with the rarefication of data towards bigger t (cf the authors' own comments about fig. 5)
  15. "a counter-intuitive effect": is this effect persistent or just happens in this particular training run / for this particular problem?
  16. "loose" -> "lose"
  17. last par on p. 10: "works ... for the uncertainty estimate": I'm not convinced (also, the authors themselves commented that the heteroscedastic uncertainty is noo large). A useful modification to fig. 3 is to show the normalised residuals: if the uncertainty quantification works, they should be normally distributed around zero with spread 1 (a histogram to the side might help to visualise this).
  18. p.11: "both computing the loss in Eq.(6)": but I thought eq (6), a modification of eq. (2) is only for the "residual" training... Again, the authors should have one definite place stating the losses they use.
  19. fig. 4: a cursory glance at figs 3 and 4 might leave the reader (me) initially thinking that the right panel of fig. 4 includes noise and residual training and is somehow magically extremely good. I'd suggest the authors put clear labels next to / on the panels of figs 3 & 4: "labelled-only, no noise", ..., "labelled + residual, no noise", etc. Also, all panels lack the "t" label for the horizontal axis..
  20. p. 12: "t = 0 < 2" probably means "0 ... 2", but I'd suggest $t \in [0, 2] \cup [7, 8]$, which is standard mathematical notation.
  21. "the poor agreement with the true solution": I wouldn't call it poor.. the green and blue lines are not wildly off and do include "reasonable" spread. Also, the "correct" behaviour (in the gaps / absence of data), again, depends on the prior, which is implicit.
  22. "uncertainty assigned but the repulsive ensembles": "but" -> "by"
  23. the final remark: which stance do the authors adopt? Indeed, do the authors believe there is a correct (i.e. mathematically provable) answer?
  24. PINNterpolation?

  25. PINNulator (I have to admit, I got a bit tired before reaching the substantive part of the paper: the application)

  26. Luminosity-distance PINN
  27. "predict ... to emulate": again, the authors should make a clear distinction between "predicting" and "emulating" and "encoding"...
  28. eq. (26): which is the network output?Is it $H$ or $d_L$?! If the latter, is the $H$ in eq (26) simply a shorthand for eq (24)? Again (and again), the authors must make clear which quantity is calculated by a neural network and what are its input. Lastly, in section 2, no mention was made of the "conditioning" variables $\lambda$, which play a crucial role in the application.
  29. eq. (26): after reaching section 4 / fig. 8, I understand this better, but consider first presenting the two approaches (H calculated from eq. (25) and learned by a NN) close together and with he help of fig. 8. However, fig. 8 (left) mentions "labelled" data, but such is not used for the PINNUlator, no?
  30. p. 14: "PNN" -> "PINN"
  31. fig.6 is extremely confusing, and should I say, misleading (to de-mislead myself, I had to read a lot of the text and check out ref 39); of course the "uncertainties" of training a network to emulate a differential equation are tiny in comparison with experimental data!! That's like saying "I solved eq. (24) using scipy.odeint and it returned $10^{-12}$ uncertainty"... This should just not be compared with data because it makes no reference to the data; whereas, the left panel of fig. 6 s suggests at first glance that the PINN is trained from that data and somehow magically extracts exactly the "true" parameters... and not to speak of how putting "data" and "truth" on the same plot implies that it's the truth for the data, which it is not! Call it "fiducial model".
  32. a propose ref 39: at first it seemed that it does _exactly what this paper presents but manages to squeeze it in an abstract... The authors should very clearly highlight the advances they c in contribute to the application of PINNs to SNaeIa in the introduction.
  33. p. 14: "away from the best fit parameters": but I thought the training was done for random $\Omega_m \in [0, 1]$, $w \in [−1.6,−0.5]$ and not simply for one set of values? Again, the "best-fit" parameters relate to Union, whereas the training has supposedly made no reference to that data.
  34. fig. 7 vs the text: the text mentions 1000 test evaluations. How are they aggregated for fig. 7? Especially, how are the 1000 individual uncertainties aggregated?

  35. PINNference

  36. "either discrete parameters or a neural network-represented free Hubble function": but scientific conclusions are usually done from results on the "discrete parameters". How do the authors foresee their method of inferring the free Hubble function be useful in science?
  37. eq. (29) "The data loss plays the same role as L_IC in Eq.(2)." Finally! This should be said much much earlier, preferably in eq. (2).
  38. fig. 8 clarifies my doubt about eq (26); I'd suggest making it more symmetric by putting, on the left panel, "parameters $\lambda$" in a separate box below "ODE and loss", in parallel with "$f_\phi(t), \sigma_ \phi(t)$" It can remain gray and with an arrow pointing towards "ODE and loss".
  39. p. 16 / fig 9 / "On this synthetic data set the Hubble function can be learned almost perfectly": was noise added to the synthetic data?
  40. p. 17, "the width of the derivative distribution is $d\sigma_\theta / dz$: I do not take this for granted. Following the discussion, $\sigma_\theta$ has absolutely no relation to $\tilde{d}_{L,\theta}$: they are simply the st.dev. and mean of a Gaussian, while $d\tilde{d}_{L,\theta} / dz$ is the variation of the mean. The authors should explain how they arrive at the above claim and be very explicit with notation: e.g. what exactly is assumed to follow the given normal distribution, the training data?
  41. eq. (31): are both $\tilde{H}$ and $\tilde{H}_\phi$ neural networks, or does the former simply stand for samples from eq. (30)?
  42. "The combination of eq (30) and (31) allows us to optimize ... simultaneously" but I thought $H_\phi$ was $f_\phi$ in eq (29) and already plotted, with uncertainty, in fig. 9? Or is it that 4.0 / fig. 9 show training simply on one odeint solution (fixed cosmology) and no noise, and so the uncertainties are just in the NN reconstruction and not due to noise, as in 4.1 & 4.2 / fig 11 ?
  43. "The sampling of ..., allowing the network parameters to influence the loss": the authors should discuss the well-known "parametrisation-trick" and the difficulties of combining sampling and gradient in general (e.g. see Black-box variational inference).
  44. 4.2
  45. "multivariate normal distribution": do the authors mean a collection of univariate normal variables, or really a correlated multivariate normal? It seems the second, but covariance at different t was never discussed in the methodological part of the paper.
  46. p. 18 "we shown" -> "we show"
  47. fig. 11: the green and orange means match pretty well, but the uncertainty of the repulsive ensemble is very small. Why? What is the interpretation of this mean orange/green line?
  48. 4.3
  49. "Naturally, this differentiation introduces a larger uncertainty": I do not see this point
  50. fig. 12:
    • "date" -> "data"
    • "propagating the error bars" is a rather strong term for the present uncertainty quantification procedure; but more importantly, I do not see uncertainties in the reconstructions!
  51. what is the interplay of the combination of the considerations "PINN is uncertain away from the initial conditions" and "dark energy has little impact on high-z"?
  52. 4.3 feels extremely rushed, whereas it is supposed to be the culmination of the whole work: an application showcasing all the bells and whistles of the work (uncertainties, repulsive ensembles...), a scientific conclusion derived from real data.... Instead, it contains none of this and is, I'm sad to say, rather disappointing.

  53. PINNclusions

  54. "emulation ... provides tremendous speedups": this should be in the intro / motivation section (as well); still, the use of a PINN emulator was not demonstrated in this work.
  55. "sensibly increasing uncertainties": I wouldn't go so far as to say "sensibly": after all, no comparison to "proper" uncertainties was ever performed

Recommendation

Ask for major revision

---

## Round 1 · Referee Report · Anonymous (Referee 3) · 2024-10-1

Report

This manuscript investigates the use of physics-informed neural networks (PINNs) to reconstruct the Hubble function, H(z), using supernova data. It builds upon related work by some of the same authors, with its main result focusing is the performance of such methods in supernova data analysis in cosmology. I consider the manuscript to be well written, although a bit excessively in a tongue-in-cheek style, and to push the field a bit further. On the other hand, I think the main science case for using PINNs in this context is not articulated clearly, and further discussions and comparisons with alternative methods are warranted. I list below in more details what I consider the major and minor issues with the manuscript, and it is my recommendation that the authors should address them before the manuscript is suitable for publication.

Major Issues

  1. The science case for using PINNs for the reconstruction of the Hubble diagram is not explored in sufficient detail. From a physics perspective, there have been a considerable number of methods proposing how to measure H(z) directly from the data without the need for standard candles, contrary to what is stated in the Introduction. To wit, using (i) radial BAO; (ii) cosmic chronometers; (iii) clustering of standard candles; (iv) the redshift drift; (v) FRBs; (vi) Alcock-Paczynski distortions. The literature on radial BAOs is extensive (e.g. arxiv:0910.5224); a review of some of the other methods is provided in arxiv:2201.07241; a proposal based on FRBs can be found in arxiv:2004.12649; one based on Alcock-Paczynski in arxiv:2406.15347. The authors should discuss how using PINNs and supernova data compares with the other methods.

  2. Besides other methods based on different data, other model-independent mathematical methods have been proposed and widely used. In particular, the most basic method to derive H(z) from distances is to employ simple binning and finite differences. Cosmographic expansions have also been widely used. More sophisticated Gaussian processes methods have also been broadly investigated recently (e.g. arxiv 1805.03595, 1907.10813, 2007.05714). Again, there is no quantitative comparison of those methods with the results from PINNs.

  3. Supernova by themselves, without Cepheids or other means of calibration, are unable to constrain H_0 as that parameter is degenerate with the absolute magnitude of the SN. Therefore they can only measure relative distances, and can only constrain H(z) up to a multiplying constant. This important point becomes implicit when you adopt dL*H_0/c as a variable below Eq. (24), but this point should be explicitly discussed. In particular, in Figure 11, the bottom panels are actually measuring H_0/H(z). The Pantheon+ collaboration also provide a Pantheon+SH0ES dataset, which include H_0 from Cepheids, and which could be used to constrain H(z) instead of H(z)/H_0.

Minor Issues

  1. The Introduction incorrectly states that the "Friedmann-Robertson-Walker spacetimes are entirely characterized by their Hubble function H(a) = adot/a". This is correct only for flat FLRW. Spatial curvature introduces physics which go beyond H(a). I.e., 2 FLRW spacetimes with the same H(a) but different curvature have different phenomenologies.

  2. "Perhaps the most direct probe of cosmic evolution out to redshifts beyond unity are supernovae of type Ia [1, 2]." Why are SN considered a more direct probe than BAO?

  3. In Section 2.3 it says "The derivation of repulsive ensembles [21, 24] starts with the usual update rule minimizing the log-probability p(θt|xtrain) by gradient descent". I guess the authors mean maximizing here, or instead minimizing (- log p).

  4. In the end of the day, the PINNs are being used to compute a derivative of a function (H from dL) based on a number of datapoints. How well can it be used to compute second derivatives, i.e. H'(z)? Presumably it would be noisier, but a quantitative analysis would benefit the manuscript and to put the results more in context.

  5. It would be interesting to include a discussion on the computational cost of training the PINN as compared to other methods such as simple Gaussian Processes.

Requested changes

The authors should address all 3 Major Issues in the report, and ideally also the 5 Minor Issues.

Recommendation

Ask for major revision

---

## Editorial Decision

resubmitted